# The Role of Reactive Oxygen Species in Acute Myeloid Leukaemia

**DOI:** 10.3390/ijms20236003

**Published:** 2019-11-28

**Authors:** Jonathan R. Sillar, Zacary P. Germon, Geoffry N. De Iuliis, Matthew D. Dun

**Affiliations:** 1Haematology Department, Calvary Mater Hospital, Newcastle, NSW 2298, Australia; 2Cancer Signalling Research Group, School of Biomedical Sciences & Pharmacy, Faculty of Health & Medicine, University of Newcastle, Callaghan, NSW 2308, Australia; Zacary.Germon@uon.edu.au; 3Priority Research Centre for Cancer Research, Innovation & Translation, Faculty of Health & Medicine, Hunter Medical Research Institute, New Lambton Heights, NSW 2305, Australia; 4Priority Research Centre for Reproductive Sciences, Faculty of Science, University of Newcastle, Callaghan, NSW 2308, Australia; geoffry.deiuliis@newcastle.edu.au

**Keywords:** acute myeloid leukaemia, reactive oxygen species, oxidative stress, NADPH oxidases, anti-oxidants

## Abstract

Acute myeloid leukaemia (AML) is an aggressive haematological malignancy with a poor overall survival. Reactive oxygen species (ROS) have been shown to be elevated in a wide range of cancers including AML. Whilst previously thought to be mere by-products of cellular metabolism, it is now clear that ROS modulate the function of signalling proteins through oxidation of critical cysteine residues. In this way, ROS have been shown to regulate normal haematopoiesis as well as promote leukaemogenesis in AML. In addition, ROS promote genomic instability by damaging DNA, which promotes chemotherapy resistance. The source of ROS in AML appears to be derived from members of the “NOX family” of NADPH oxidases. Most studies link NOX-derived ROS to activating mutations in the Fms-like tyrosine kinase 3 (FLT3) and Ras-related C3 botulinum toxin substrate (Ras). Targeting ROS through either ROS induction or ROS inhibition provides a novel therapeutic target in AML. In this review, we summarise the role of ROS in normal haematopoiesis and in AML. We also explore the current treatments that modulate ROS levels in AML and discuss emerging drug targets based on pre-clinical work.

## 1. Acute Myeloid Leukaemia

Acute myeloid leukaemia (AML) is a highly aggressive heterogeneous cancer of immature cells of the myeloid lineage known as myeloblasts, or AML blasts. Recurring somatic mutations in epigenetic modifying genes block differentiation into more mature forms, leading to clonal expansion within the bone marrow and blood. This leads to bone marrow failure and ultimately death without successful treatment. Typically, patients present with clinical symptoms related to failure of normal haematopoiesis including fatigue, breathlessness, bleeding events, and infections. Overall survival has improved marginally over the last 40 years, predominantly thanks to improvements in supportive care. Most of the gains, however, are in seen younger patients (<40 years) who have an estimated 5 year overall survival of ~50%. Outcomes for older patients’ remain dismal with less than 5% of patients over the age 70 surviving long term [1].

In recent years, the genomic landscape of AML has been well characterized and has highlighted the biological heterogeneity of AML [2,3]. The treatment of AML, however, has remained largely unchanged since the advent of the so-called “7 + 3” regimen in the 1970s, with younger fit patients being transplanted once they reach their first complete remission, apart from patients deemed to have a favourable genetic profile, and older patients receiving palliative chemotherapy and transfusion support only. In the last two years, the United States Food and Drug Administration (FDA) has approved a number of novel targeted therapies for AML. These include the small molecule inhibitor Rydapt (midostaurin) for Fms-like tyrosine kinase 3 (FLT3)-mutant patients [4], and Tibsovo (ivosidenib) [5] and Idhifa (enasidenib) [6] for isocitrate dehydrogenase (*IDH1* and *IDH2*)-mutant patients [7], as well as agnostic therapies such as the B-cell lymphoma 2 (*BCL2*) inhibitor Venetoclax [8]. Whilst responses appear impressive, relapse rates remain high and follow-up durations are short, highlighting the importance of ongoing research investigating the underlying biological processes that drive AML initiation and progression [9]. This review outlines the emerging roles reactive oxygen species (ROS) play in the pathogenesis of AML, with a focus on current and future therapeutic interventions where modulating ROS levels may play a key role.

## 2. Overview of Reactive Oxygen Species and Redox Homeostasis

ROS are a heterogeneous group of molecules and radicals including superoxide anion (O_2_^•−^), hydrogen peroxide (H_2_O_2_), and the hydroxyl radical (HO^•−^). ROS have been extensively studied in cancer over the last 20 years [10] and have been implicated in neurodegeneration [11,12], ageing [12,13], diabetes [14,15]**,** and hypertension [16]. Mitochondria were first discovered to produce ROS, namely hydrogen peroxide, over fifty years ago [17]. Whilst previously it was thought that these ROS were non-functional by-products of cellular metabolism causing tissue damage and promoting disease through lipid peroxidation and DNA damage, it is now well recognised that they play an important role in cellular signalling in both physiological and pathological cellular processes [18].

Superoxide anion is produced by a one electron reduction of oxygen and is the precursor to most ROS. During oxidative phosphorylation, 1–5% of electrons escape from the electron transport chain, giving rise to a background level of superoxide in the presence of oxygen [19]. It has been shown in vitro that superoxide is predominantly produced from Complexes I and III of the electron transport chain, with some variability depending on cell type [20,21,22]. Given the physiological amounts generated, superoxide can be dismutated to hydrogen peroxide, which can be further reduced to water via the cooperation of superoxide dismutase and catalase; however, some superoxide can be partially reduced to the more damaging hydroxyl radical [21]. Reactive nitrogen species (RNS), the most proximal of which is peroxynitrite, are generated when superoxide reacts with nitric oxide (NO), which is generated by nitric oxide synthases [23]. RNS will not be directly addressed in this review, but it is possible that they cooperate with ROS in regulating haematopoiesis as well as driving leukaemogenesis in AML, although the evidence is currently limited.

The phagocyte nicotinamide adenine dinucleotide phosphate (NADPH) oxidase (NOX) was the first enzyme system found to produce ROS not merely as a by-product of cellular metabolism, thus suggesting a more profound role for ROS within biological systems [24]. Initially, thought to be limited to phagocytes, NADPH oxidases have now been found in almost every tissue and are involved in a wide range of physiological processes including host defence, post-translational modifications of proteins, cellular signalling, regulation of gene expression, and cell differentiation [24]. Collectively they are now known as the “NOX family” of NADPH oxidases and include NOX1 (*NOX1*), NOX2 (*CYBB*), NOX3 (*NOX3*), NOX4 (*NOX4*), NOX5 (*NOX5*), dual oxidase 1 (*DUOX1*), and DUOX2 (*DUOX2*)—NOX2/*CYBB* (also known as gp91^phox^) being the originally described as phagocyte NADPH oxidase. All NOX family members are transmembrane proteins that utilise intracellular NADPH to reduce extracellular oxygen to ROS, by effectively transporting electrons across the membrane [24]. NOX1–4 require the close interaction with p22^phox^ (*CYBA*) for electron transfer [25], whilst NOX5 and DUOX1 and 2 contain EF hands that are calcium-dependent [26]. NOX1–5 produce the superoxide anion whilst DUOX1 and 2 (and possibly NOX4) produce hydrogen peroxide, as they contain a peroxidase-like domain [27]. Each NOX family member also requires a number of regulatory subunits for activation. NOX2, as an example, is regulated by p40^phox^ (*NCF4*), p47^phox^ (*NCF1*), p67^phox^ (*NCF2*), and the Ras-related C3 botulinum toxin substrate (*RAC1*) that are found in the cytoplasm [26]. This complex interaction involves both the exchange of GDP for GTP on Rac and the phosphorylation of the p47^phox^ subunit, which allows p47^phox^ to bind with the p22^phox^/NOX2 complex (vesicle bound), leading to the subsequent fusion of NOX2 containing vesicles with the phagosomal membrane [24]. The activated phagocyte NADPH oxidase then is able to generate the ‘respiratory burst’ required for destroying microbes.

A number of other metabolic sources of ROS generation have been discovered. Cytochrome P450 is the terminal oxidase of the membrane-bound microsomal monooxygenase system (MMO), which is localized in the endoplasmic reticulum and is primarily found in the liver [28] as a detoxification enzyme. The main function of MMO is to oxidise exogenous compounds, in most cases leading to oxygenation of the substrate, to facilitate its excretion from the body. This oxidative process commonly results in the production of ROS as a by-product, and similar to oxidative phosphorylation, the ROS from MMO also appear to play an important role in cell signalling [29]. Xanthine oxidase, which catalyses the oxidation of hypoxanthine to xanthine and further xanthine to uric acid, results in ROS production. Xanthine oxidase is a key enzyme of the peroxisome system, which has been shown to produce both ROS and RNS. Toward regulating states of oxidative stress, peroxisomes also contain a number of ROS scavenging enzymes including catalase and glutathione peroxidase [30]. This highlights a subset of the complexity regarding intracellular ROS production as well as the wide spectrum of subcellular locations in which redox homeostasis appears important to many cellular functions.

Given the reactivity of ROS and therefore their potential harmful effects on cells and tissue, redox homeostasis is tightly regulated with oxidative stress largely prevented via a suite of anti-oxidant systems. The dismutation of superoxide, for example, can occur spontaneously (particularly at a low pH) but is predominantly regulated enzymatically via a mitochondrial specific superoxide dismutase (now named SOD2) to hydrogen peroxide [31]. SOD isoforms, SOD1 and SOD3, perform the same function in the cytoplasm and extracellular space respectively. Hydrogen peroxide is more stable than superoxide, has a longer half-life and importantly, can readily diffuse across cellular membranes [32] unlike the superoxide anion. Interestingly, SOD2 knockout mice show lethality in the first week of life through the accumulation of oxidative DNA damage caused by superoxide [33]. Furthermore, factors known to cause oxidative stress, such as ionising radiation and hyperoxia, have been shown to induce SOD2 expression via the activation of the nuclear transcription factor NFkB [34,35], which increases the clearance of superoxide from the mitochondrial intermembrane space. SOD2 can achieve a net ROS reduction in the mitochondrial space by converting the membrane-impermeable superoxide anion to hydrogen peroxide, which is then free to diffuse into the cytoplasm. The conversion of superoxide to hydrogen peroxide is, thus, essential for intracellular redox homeostasis.

Whilst hydrogen peroxide is considered relatively stable compared to other ROS, it can also lead to oxidative damage to lipids and proteins. This is particularly so in the presence of iron, where hydrogen peroxide is readily converted to hydroxyl radicals via Fenton chemistry. A number of anti-oxidant systems, therefore, regulate cytoplasmic hydrogen peroxide levels. Glutathione (GSH) is a tripeptide (composed of cysteine, glutamic acid, and glycine) and is present in all mammalian tissues (reviewed in [36]). It is the most abundant non-protein thiol and is biologically present at 1–10 mM concentrations, with the highest concentrations found in the liver (up to 10 mM). GSH exists in two forms, the thiol-reduced form and the disulphide-oxidised form (GSSG), with >98% existing in the reduced form, thereby providing an important source of electrons for anti-oxidant processes. It is found within different subcellular compartments including the cytosol, mitochondria, endoplasmic reticulum, and nucleus, as well as within extracellular fluid [37]. Its key role as a regulator of oxidative stress is via the reduction of hydrogen peroxide to water and oxygen. This redox reaction is catalysed by the GSH-peroxidase (GPx) and peroxiredoxin (PRDX) families [37]. The pool of GSH is utilised by GSH peroxidase (GPx)-catalysed reactions that reduce ROS, forming oxidised GSSG, which in turn are reduced back to GSH by GSSG reductase at the expense of NADPH, and thus the reserve of reduced GSH is maintained [36]. Catalase enzymes are another important regulator of intracellular ROS. They exclusively convert hydrogen peroxide to oxygen and water and are found primarily in the cytosol and peroxisomes [38,39]. Similar to GSH-supported anti-oxidant systems, catalase diminishes the ROS pool since its reaction products are largely redox-inert. Peroxiredoxins (Prdxs) are a family of anti-oxidant proteins that contain a conserved peroxidatic cysteine (Cp), which reacts with peroxides to form cysteine sulfenic acid (Figure 1). This then reacts with another cysteine residue, named the resolving cysteine (Cr), to form a disulphide bond that, in turn, can be reduced by an electron donor, such as GSH or thioredoxins (Trxs) [40].

## 3. The Role of ROS in Normal Haematopoiesis

There is significant evidence to support the role of ROS in regulating haematopoiesis (Figure 2). Haemopoietic stem cells (HSCs) have the capacity for both self-renewal (give rise to identical daughter cells) as well as being able to differentiate into all ten blood lineages: erythrocytes, platelets, neutrophils, eosinophils, basophils, monocytes, T and B lymphocytes, natural killer cells, and dendritic cells. They are found in relatively hypoxic environments within the bone marrow, known as osteoblastic niches [41]. An *in vitro* study of HSCs isolated from mouse bone marrow samples cultured in 1% oxygen suggested that a hypoxic environment inhibited proliferation and thus favoured quiescence in HSCs [42]. This appeared to be mediated by increased expression of hypoxia inducible factor (HIF) 1 alpha (*HIF1A*), a key transcriptional regulator of cellular responses to oxygen variation [43] (Figure 2). The knockdown of *HIF1A* and *HIF2A* has been shown to impede the long-term repopulating ability of human CD34+ cord blood cells via increased ROS production [44].

The FoxO (Forkhead) family of transcription factors has also been shown to regulate HSC self-renewal and survival (Figure 2). FoxO-deficient HSCs (*FoxO1/3/4^−/−^*) display defective long-term repopulating activity correlating with increased cell cycling and apoptosis [45]. These changes were associated with a marked increase in ROS as compared to wild-type HSCs. Anti-oxidant therapy with *N*-acetyl-l-cysteine (NAC) was able to reverse this phenotype. This work was supported by Miyamoto et al., who showed that in *FoxO3a^−/−^* HSCs there was defective maintenance of quiescence with an associated increase in ROS as well as increased phosphorylation of p38 mitogen-activated protein kinase—p38MAPK (*Mapk14*) (involved in apoptotic pathways) [46] (Figure 2). Ataxia telangiectasia mutated (*Atm^−^*^/*−*^) knockout mice (proposed as a model for oxidative stress in HSCs) developed progressive bone marrow failure in another study due to a defect in HSC function mediated by increased ROS that was again rescued by treatment with NAC [47]. The tuberous sclerosis complex (TSC)-mammalian target of the rapamycin (mTOR) pathway is a key regulator of cellular metabolism. It was reported that TSC1 deletion resulted in increased ROS levels in HSCs driving them from quiescence to rapid cell cycling. Furthermore, reduced haematopoiesis and self-renewal of HSCs was observed [48]. These studies taken together highlight the importance of a low ROS environment in order to maintain HSC quiescence, self-renewal, and long-term survival. They also suggest a number of potential regulators of ROS in HSCs including FoxO, Atm, and the TSC-mTOR pathway (Figure 2).

A seminal study by Jang and Sharkis identified two differing murine HSC populations designated as ROS^low^ and ROS^high^ [49]. Both populations expressed the same surface markers but displayed markedly different phenotypes. The ROS^low^ population displayed less activation with greater self-renewal capacity and reduced phosphorylation of p38-MAPK (i.e., reduced apoptosis). The ROS^high^ population displayed the opposite phenotype [49] (Figure 2). Using a serial transplantation model, the authors were also able to demonstrate that ROS^low^ cells had a greater repopulating capacity than ROS^high^ cells suggesting enrichment for long-term HSC within the ROS^low^ compartment. Beyond that, the ROS^low^ cells showed a greater expression of calcium-sensing receptors (CaRs) and N-Cadherin, both of which have been shown in previous studies to be found on HSCs residing within the osteoblastic niche [50,51,52] (Figure 2). CaRs are important for HSC localisation to the osteoblastic niche, whereas N-Cadherin, in conjunction with calcium ions, adheres HSCs to the niche. An anti-cancer drug, 5-flourouracil, was shown to increase ROS in HSCs and decrease N-Cadherin expression, thus detaching them from the osteoblastic niche [53]. Therefore, it appears that the ROS^low^ cells reside in a relatively quiescent state in the osteoblastic niche, whereas the ROS^high^ cells reside in the vascular niche, adjacent to peripheral blood, whereby they undergo differentiation (Figure 2).

There is evidence that ROS prime myeloid progenitors to differentiate. In vivo studies of *Drosophilia*, multipotent haematopoietic progenitors (functionally akin to common myeloid progenitors) have shown that increasing ROS prime these cells for differentiation into more mature myeloid cells. ROS scavengers were shown to retard differentiation, whilst mutating SOD2 was shown to increase the number of differentiated cells [54]. This is concordant with the work by Tothova et al., where FoxO knockout mice displayed increased ROS in HSCs leading to myeloid differentiation [45] (Figure 2). In mammalian cells, ROS have also been shown to regulate myeloid differentiation. Megakaryocyte differentiation into mature platelets, for example, has been shown to be regulated by ROS [55,56,57]. In addition, mitochondrially derived ROS have been shown to trigger haematopoietic stem cell differentiation through NOTCH1 degradation by autophagy in a murine model [58].

In summary, these studies highlight the important role that ROS play in maintaining HSC quiescence, self-renewal, and long-term survival (Figure 2). In addition, they outline the role that ROS plays in differentiation from haematopoietic stem cells to terminally differentiated myeloid progenitors. What these studies also highlight is the sensitivity of HSCs to damage caused by oxidative stress and the profound degree to which ROS levels must be regulated in HSCs and early progenitors.

As previously discussed, the “NOX family” of NADPH oxidases play a key role in intracellular ROS generation. Within haematopoietic stem cells, the expression of NOX isoforms has not been uniformly characterised. Piccoli et al. identified NOX1, 2, and 4 as well as their corresponding regulatory subunits in CD34+ haemopoietic stem cells [59]. This is at odds with the work of Sanchez-Sanchez et al., who identified all NOX isoforms in CD34+ cells except NOX 1 and 4. Their work further identified NOX2 and NOX5 expression in CD33+ myeloid progenitors, whereas mature CD15+ myeloid cells expressed NOX2, NOX5, DUOX1, and DUOX2 suggesting that different NOX isoforms may be functionally important at varying stages of myeloid cell maturation [60]. NOX2 is expressed on mature granulocytes and upon stimulation generates the respiratory burst required for killing phagocytised microorganisms. Chronic granulomatous disease is caused by genetic mutations in any of the four regulatory subunits of NOX2 and results in increased susceptibility to bacterial and fungal infections, thus exemplifying the importance of this enzyme system to neutrophil function [61]. NOX2 is universally expressed on macrophages (tissue-specific phagocytes), but there is also evidence of NOX4 expression [24,62]. Eosinophils, dendritic cells, and platelets all express NOX2 [63,64]. NOX2-derived ROS maintains an alkaline pH in the phagasome of dendritic cells, which impairs antigen degradation, thus enhancing cross-presentation. Dendritic cells lacking NOX2 have an impaired cross-presentation of antigens to T lymphocytes as a consequence of increased antigen degradation [63]. Therefore, in summary, there is evidence of expression of ‘NOX family’ members in myeloid cells ranging from haemopoietic stem cells through to terminally differentiated cells of the myeloid lineage, with evidence in some cases of functional consequences in knockout models and patients.

## 4. ROS in AML

In recent years, it has become increasingly apparent that ROS are elevated in cancer cells [10] (Figure 3). In myeloid neoplasms there is evidence for increased ROS in myelodysplastic syndrome [65], chronic myelomonocytic leukaemia [66], chronic myeloid leukaemia [67]**,** and myeloproliferative neoplasms [68]. This is no different in AML, where the roles for ROS in driving leukaemogenesis are ever expanding. Direct evidence for elevated ROS levels compared to normal myeloblasts comes from a number of studies. Hole et al., identified increased extracellular superoxide production in 65% of primary AML blasts compared with controls [69]. In some samples, 100-fold greater levels were measured, however, there did not appear to be any correlation with the underlying molecular subtype of AML. A number of other studies have focused on the role of mutant receptor kinases in driving ROS production in AML. Ras mutations occur in approximately 10–15% of AML cases and do appear to have a significant impact on prognosis [70]. In further work by Hole et al., they used murine (Sca+) and human (CD34+) myeloid progenitor cells retro-virally transduced with *H-Ras* and *N-Ras* as a model to study ROS [71]. Activated Ras promoted increased ROS production as well as growth factor independent proliferation without alteration in anti-oxidant expression. A murine *K-Ras* myeloproliferative disease model was also shown to drive increased levels of ROS [72].

Mutations of the Fms-like tyrosine kinase 3 (*FLT3*) receptor occur in approximately 30–35% of AML [2,9]. These mutations result in constitutive activation of the receptor in the absence of the FLT3 ligand and include internal tandem duplications (ITDs) of the juxtamembrane domain [73] and, less commonly, point mutations within the tyrosine kinase domain (TKD) [9]. FLT3 mutations are now strongly implicated in driving increased ROS production in AML. A number of studies using the mouse haematopoietic progenitor cell line 32D transduced to stably express human FLT3-ITD mutations, produce high levels of endogenous ROS and increased oxidative DNA damage compared to parental cells [74,75]. Additionally, similar findings were seen in Ba/F3 mouse progenitor cell lines transfected with both the FLT3-ITD and FLT3-TKD (D835Y) constructs compared to parental cells, as well as in primary AML cell lines with the FLT3-ITD mutation (MV4-11, MOLM-13) compared to a wild-type FLT3 cell line REH. This was associated with evidence of increased DNA damage, suggesting a mechanism by which FLT3-mutant AML drives genomic instability through increased ROS production. FLT3 inhibition with CEP-701 resulted in reduced cellular ROS levels and a decrease in the number of double-stranded DNA breaks. FLT3 inhibition with PKC412 reduced ROS levels and double-stranded DNA breaks. Furthermore, oxygen consumption rates have been shown to be higher in cells harbouring BCR-ABL, JAK2 V617F, and FLT3-ITD mutations than controls, abrogated upon inhibition with tyrosine kinase inhibitors [76]. Hence, it is clear that oncogenic tyrosine kinases, including the FLT3-ITD and FLT3-TKD mutations, as well as Ras GTPase mutations, drive increased intracellular ROS levels in AML (Figure 3).

Another interesting subset of AML patients includes those harbouring mutations in the isocitrate dehydrogenase genes (*IDH1* and *IDH2*), seen in 10–15% of cases [2]. These mutations lead to the production of an oncometabolite known as R-2-hydroxyglutarate (R-2HG), which blocks myeloid differentiation through epigenetic modulation. Interestingly, there is also evidence that R-2HG increases intracellular ROS, which, in turn, via an extracellular signal-regulated kinase (ERK)-dependent pathway, phosphorylates NF-kB and stimulates proliferation of AML cells [77]. A study using primary AML bone marrow samples compared ROS levels across molecular and cytogenetic subtypes [78]. They found FLT3-ITD-positive and core binding factor AML samples had higher intracellular ROS levels compared to FLT3-ITD-negative samples. Samples with nucleophosmin (NPM1) mutations, which occur in up to 30% of AML cases, appeared to have lower levels of ROS even in the presence of co-existing FLT3-ITD mutations. This is interesting based on clinical outcome data, which clearly demonstrates that FLT3-ITD-mutant AML with a co-existing NPM1 mutations have a more favourable prognosis that those with FLT3-ITD mutations and wild-type NPM1. The ROS production in AML promotes second messenger signalling and drives transcription (STAT5), increased DNA damage, and lipid peroxidation (Figure 3).

### 4.1. NOX Family Enzymes in AML

As discussed in Section 2, the primary intracellular sources of ROS production are the mitochondria and the “NOX family” of NADPH oxidases. The source of ROS within AML has been explored in a number of studies and almost uniformly point to the ‘NOX family’ as the key source. In the aforementioned study by Hole et al., higher levels of extracellular superoxide were observed in primary AML blasts compared to normal bone marrow samples. In contrast, the AML samples had lower levels of mitochondrial ROS. Further, NOX inhibitors significantly reduced superoxide generation as opposed to electron transport chain inhibitors and mitochondrial ROS scavengers, which had little or no impact on ROS levels, implicating NOX as the primary source of ROS in AML [69]. This work is supported by a number of other studies showing significant reductions in intracellular ROS levels in vitro upon NOX inhibition or knockout of NOX isoforms or subunits [75,79,80,81]. In contrast to this, Reddy et al. did not observe any change in ROS levels after NOX2, NOX4, and p22^phox^ knockdown in an FLT3-ITD AML cell line (MOLM-13), although this does not seem to have been replicated in other studies [76]. There is also conjecture as to the specific NOX isoform present in AML. In primary AML blasts, NOX1, NOX2, and NOX4 gene and/or protein expression have been identified [69,81]. In AML cell lines, gene and/or protein expression of NOX2, NOX4, NOX5, p22^phox^, p40^phox^, p47^phox^, and p67^phox^ have all been reported [75,76,80,81]. Knockdown of p22^phox^ and NOX4 have been shown to reduce levels of ROS in AML cell lines [75,81]. Furthermore, ROS production was completely abolished in a *Nox2^−/−^* mouse model of Ras-activated Cd34+ progenitor cells [71]. Hence, the data is somewhat conflicting, with the strongest evidence supporting NOX2 and NOX4.

### 4.2. Anti-Oxidants in AML

There are a number of studies reporting dysregulation of anti-oxidants in AML. One of the earliest studies indirectly linking ROS to AML pathogenesis reported that SOD2 levels were reduced in AML cells as compared to normal granulocytes [82]. A recent study compared blood levels of oxidative stress markers and anti-oxidant level in healthy volunteers and patients with acute lymphoblastic leukaemia (ALL) and AML. Interestingly, they also showed reduced levels of SOD, glutathione, and catalase compared to healthy controls, with an expected increase in malondialdehyde, a well-defined marker of oxidative stress [83]. Another study demonstrated increased levels of oxidised glutathione in CD34+ AML cells compared to normal bone marrow samples [84]. A proteomic analysis of primary AML blasts observed increased protein expression of catalase and peroxiredoxin-2 with some variability across FAB subtypes [85]. In acute promyelocytic leukaemia (APL) cell lines, increased catalase expression has been shown to correlate with arsenic trioxide (ATO) resistance, consistent with ATO’s known ROS-dependent cytotoxicity [86]. Therefore, it would appear that there is significant dysregulation of anti-oxidant protein expression in AML compared to normal controls. The observed increased catalase and peroxiredoxin expression would confer a protective mechanism to cells with high levels of ROS production, whereas lower levels of SOD expression would lead to increased superoxide levels and lead to intracellular oxidative stress. This stress, in turn, may create genomic instability and promote resistance to therapy. The differential expression between disease state and normal provides a therapeutic opportunity as will be discussed in Section 5.

### 4.3. ROS-Regulated Second Messenger Signalling in AML

#### 4.3.1. Phosphatases

The emergent role of ROS in second messenger signalling has prompted research into the pathways affected in many cancers, including AML. ROS have been shown to regulate protein function via oxidation of the thiol functional groups in cysteine residues [87] (Figure 1). There is a wide range of known oxidative post-translational modifications (reviewed in [88]), which ultimately lead to alteration of protein structure and function (Figure 1). Protein tyrosine phosphatases (PTPs), for instance, contain a conserved cysteine residue in their catalytic domain, which through reversible oxidation can lead to phosphatase inactivation, halting its capacity to dephosphorylate target proteins [89]. In FLT3-ITD+ AML, there is evidence that NOX4-derived ROS lead to oxidative inactivation of protein tyrosine phosphatase, receptor J—PTPRJ (*PTPRJ*) (Table 1), and that NOX4 knockdown restored phosphatase activity [81]. PTPRJ has been shown to directly interact with the FLT3 receptor and to negatively regulate its function. PTPRJ depletion leads to increased phosphorylation of FLT3 and enhanced activation of ERK and cellular proliferation [90] (Figure 3). Although not reported in AML, NOX5-derived ROS have been demonstrated to lead to inactivation of Src homology region 2 domain-containing phosphatase 1—SHP1 (*PTPN6*) (Table 1), another PTP, in a model of hairy cell leukaemia [91]. SHP1 negatively regulates FLT3 signalling, and it is possible that a similar mechanism exists in AML.

The protein phosphatase 2A—PP2A—is a serine/threonine phosphatase inactivated in many cancers including AML [92]. PP2A inhibition is essential for leukaemias driven by oncogenic mutant c-KIT [93] and FLT3 [94], with the PP2A-C catalytic subunit (*PPP2CA*) oxidised at cysteine residues, Cys266/269, contributing to the loss of the phosphatase activity under oxidative stress [95]. ROS-driven nitration of Tyr289 on PP2A-B56δ subunits (*PPP2R2B*) has also been shown in Jurkat cells and clinical human lymphomas to inhibit PP2A holoenzyme assembly and leading to enhanced S70 phosphorylation of Bcl-2 and apoptotic resistance to anticancer drugs [96].

#### 4.3.2. Kinases and GTPases

Studies on signalling pathways regulated by ROS have primarily explored those downstream of oncogenic tyrosine kinases including FLT3 and the Ras-GTPases. NOX inhibition and *p22^phox^* knockout in FLT3-ITD models of AML lead to reduced phosphorylation of signal transducer and activator of transcription 5 (*STAT5*), a master transcriptional regulator downstream of FLT3, accompanied by reduced cellular growth and migration [79,80]. BCR-ABL-driven chronic myeloid leukaemia (CML) and Janus kinase 2 (*JAK2*) V617F-mutated myeloproliferative neoplasm (MPN) cell lines showed similar results, suggesting a role for NOX-derived ROS in myeloid neoplasms driven by oncogenic tyrosine kinases more broadly [76]. Phosphorylated STAT5 has also been shown to co-localise with Rac1, an activating component of the NOX complex, suggesting a mechanism in which FLT3-ITD through phosphorylated STAT5 generates ROS from the NOX complex [74]. An autocrine mechanism of increased ROS production also appears to potentiate the oncogenic signalling capacity of FLT3-ITD. Emerging data shows that ROS themselves drive the activity of wild-type and mutant FLT3 through oxidation of cys790 in FLT3 (Table 1) changing intramolecular interactions and promoting the activity of STAT5 [101]. This may drive altered gene expression through the regulation of a number of other transcription factors as a response to intracellular oxidative stress (Figure 3). This clearly makes sense as an adaptive mechanism. Although not specifically reported in AML, the redox regulation of nuclear factor kappa B (NFkB), redox-factor 1 (Ref-1), activator protein 1 (AP-1), p53, and hypoxia inducible factor 1-alpha (HIF-1α) have all been described (reviewed in [18,108]). It is possible that increased gene expression of anti-oxidant proteins modulated by elevated ROS, while facilitating genomic instability, also confers a survival advantage in AML.

More directly, NOX2-derived ROS drive oxidation of active-site cysteines in tyrosine kinases such as the epidermal growth factor receptor (*EGFR*) (Table 1) and enhance their activity [99]. Although the pharmacological inhibition of EGFR plays no role in the treatment of AML, chronic exposure to oxidative stress drives resistance to EGFR-targeted therapies such as afatinib [100], providing a mechanism by which ROS promote resistance to targeted therapies. Relevant to leukaemia, ten other kinases harbouring catalytic cysteines localised in similar positions as the active-site cysteine 797 in EGFR, include the tyrosine-protein kinases BTK and JAK3 [109]. Pharmacologic targeting of BTK using ibrutinib inhibits AML blast proliferation and significantly augments cytotoxic activities of standard of care AML chemotherapies; cytarabine, or daunorubicin [110]. Interestingly, murine neutrophils lacking BTK (*Btk^−/−^*) produce more ROS and show hyperphosphorylation and activation of phosphatidylinositol-3-OH kinase (PI3K) and protein tyrosine kinases (PTKs), a phenotype driven by NOX2 [111]. Whether these hyperactivated kinases also harboured high-levels of cysteine oxidation driving their activity remains to be determined. Mutations to *JAK3* in myeloid [112] and lymphoid malignancies [113] drive the activity of the STAT5 signalling axes. However, unlike what has been shown in tyrosine kinases with conserved cysteine localised analogues to EGFR, oxidation of JAK3 (and JAK2) inhibits their autokinase activity (Table 1), a phenotype rescued using reducing agents [102]. As mentioned in Section 4.3.2, inhibition of NOX in FLT3-ITD AML cells suppresses STAT5 and oncogenic transcription to repress growth and survival, whereas mediators of oxidative stress such as nitric oxide and thiol redox reagents, through oxidation of crucial dithiols to disulphides within JAK2/3, inhibit interleukin 3-triggered in vivo activation, a phenomenon that is correlated with inhibited proliferation of lymphoid cells. This highlights the complexities of how oxidative stress regulates the activity of key signalling proteins in AML, with what seems to be enzyme specific regulation. Whether oxidative stress drives oxidation and activity of kinases such as EGFR / JAK in AML is yet to be determined; however, the observation of increased kinase activity and synergy following loss of function of BTK provides us with further clues into the complexities of signalling at a systems-based level.

### 4.4. ROS in the Microenvironment of AML

Modulation of the tumour microenvironment by ROS has also been described in AML. NOX-derived ROS from tumour infiltrating macrophages/monocytes have been shown to impair the function of T cells and natural killer (NK) cells. This process can be reversed with histamine that acts on H2 receptors to indirectly inhibit NOX activity [114]. In a Phase 3 clinical trial, leukaemia-free survival was observed in patients receiving interleukin 2 and histamine dihydrochloride as consolidation therapy for AML [115]. The proposed mechanism suggests that by reducing ROS there is improved functionality of the NK and T cells in the tumour microenvironment promoting activity against residual leukaemic cells that are present post-intensive induction chemotherapy. Another interesting pre-clinical study suggests a novel mechanism by which the NOX2-derived superoxide from AML blasts leads to the bone marrow stromal cell-to-AML transfer of mitochondria via nanotubes [116] (Figure 3). NOX2 inhibition was shown to reverse this process and lead to improved mouse survival in a murine model of leukaemia. Given the reliance of AML blasts on oxidative phosphorylation to generate ATP, this then confers a survival advantage. Therefore, ROS derived from AML blasts appear to not only regulate intracellular signalling pathways but also appear to modulate cells in the microenvironment in a paracrine manner to promote leukaemogenesis

## 5. ROS Modulation as a Therapeutic Target in AML?

Given the evidence for ROS promoting leukaemogenesis in AML, we sought to review the current support for modulation of ROS as a therapeutic intervention. Inadvertently, increasing ROS levels have been used therapeutically for some time now with traditional chemotherapeutic agents. Given the higher levels of ROS associated with AML, the hypothesis of increasing ROS to ‘tip leukaemic blasts over the edge’ makes sense. In APL, ATO has been shown to both increase intracellular ROS production and gene expression of proteins comprising the NOX2 subunit [117]. Further, synergistic cytotoxicity has been observed by combining ATO with phorbol myristate acetate (PMA), a known activator of NOX2. ATO-derived ROS have also been shown to promote degradation of the PML-RARA fusion protein [118]. Both cytarabine and daunorubicin, part of the standard ‘7 + 3’ induction regimen for AML, have been shown to induce increased ROS production in cell line models of AML [119]. In addition, many experimental drugs have been tested in pre-clinical models of AML, in which ROS are implicated in promoting apoptosis and cell death [120,121,122,123,124]. The downside of ROS-inducing agents is the potential for creating genomic instability and driving resistance [125], as well as oxidative damage to bystanding cells, with complications including cardiac and neurotoxicity.

The alternative approach is to use agents that inhibit ROS production or function as ROS scavengers in treating AML. Using anti-oxidants to prevent or treat cancer is not a new concept. There are vast numbers of trials that have explored the use of anti-oxidants in cancer prevention and treatment. Despite this, a systematic review failed to demonstrate a benefit for anti-oxidant supplementation in preventing cardiovascular disease and cancer [126]. In addition, there is some evidence that anti-oxidant therapy actually promotes cancer progression when combined with a standard of care chemotherapeutics in the treatment of various cancers [127]. In the case of AML, there does not appear to be any literature in which anti-oxidants supplements have been specifically used to treat AML alone or in combination with standard therapies. Treatments targeting the source of ROS production have not been reported in clinical trials. Indirectly, histamine dihyodrochloride was used to reduce ROS secretion into the tumour microenvironment as a means of reactivating NK and T cells as previously described [115]. Drugs inhibiting oxidative phosphorylation, such as metformin [128,129] and tigecycline [130,131], have been explored in preclinical models and human trials of AML without any great success to date, as the data would suggest that the NOX family are the primary drivers of ROS production in AML, which may partly explain this. Given that the primary function of the NOX family is to produce ROS, any clinical benefit of NOX inhibitors can be more readily attributed to decreasing ROS levels. There have been no reported clinical trials with NOX inhibitors used in AML despite emerging pre-clinical evidence that would suggest a potential benefit. This is likely due to the lack of isoform-specific NOX inhibitors with good in vivo pharmacokinetic data available [132]. Nevertheless, targeting NOX2 appears promising, particularly in FLT3-mutant AML, based on pre-clinical in vitro data available; however, ongoing efforts are required to assess if this will translate into a meaningful clinical benefit.

## 6. Concluding Remarks

There is now conclusive evidence that ROS are elevated in AML. It is likely that there are multiple sources of ROS including the mitochondria, NOX complex, and other metabolic sources; however, most evidence to date supports the NOX complex, particular NOX2, as the primary driver of ROS production in AML. ROS appear to confer a survival advantage to leukaemic blasts through modulation of oncogenic signalling pathways via the oxidation of critical cysteine residues in key proteins such as protein tyrosine phosphatases, oncogenic kinases, and anti-oxidant proteins (Table 1 and Figure 3). Targeting ROS in AML have already proven effective with ROS induction through the actions of chemotherapeutic agents and novel drugs such as arsenic trioxide, despite the potential for creating treatment-resistant AML clones. ROS scavengers and anti-oxidants have not been proven effective in preventing or treating cancer, and there is no evidence to support this approach in AML. Targeting the drivers of ROS production in AML is yet to be tested in vivo but provides a novel therapeutic approach with promising pre-clinical data.

## Figures and Tables

**Figure 1 ijms-20-06003-f001:**
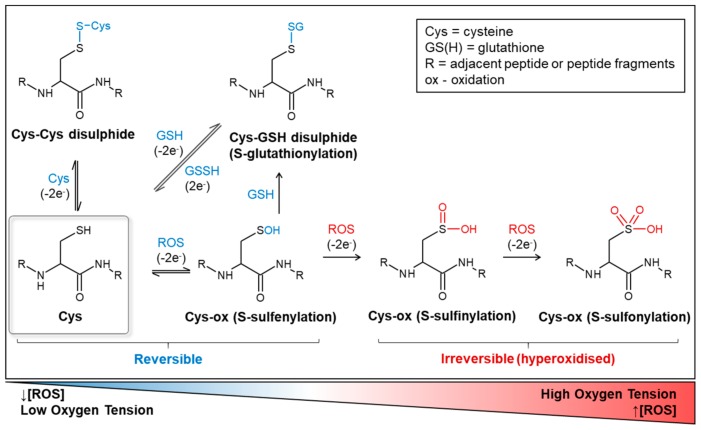
Redox-mediated post-translational cysteine modifications. Cysteine post-translational modifications are a key mechanism by which protein activity can be modulated or influenced. Oxidative cysteine thiol modifications can be broadly grouped into reversible and irreversible modifications. The reversible modifications, including the formation of the sulfeinc acid via s-sulfenylation, and the familiar formation of disulphide bridges, provide a mechanism to sense local redox states. While some redox-sensitive proteins can be activated by these modifications (e.g., EGFR), others can be silenced (e.g., PTPs). Most irreversible modifications, largely lead to a loss of function and are a result of multiple thiol oxidation steps that can occur under overt oxidative stress.

**Figure 2 ijms-20-06003-f002:**
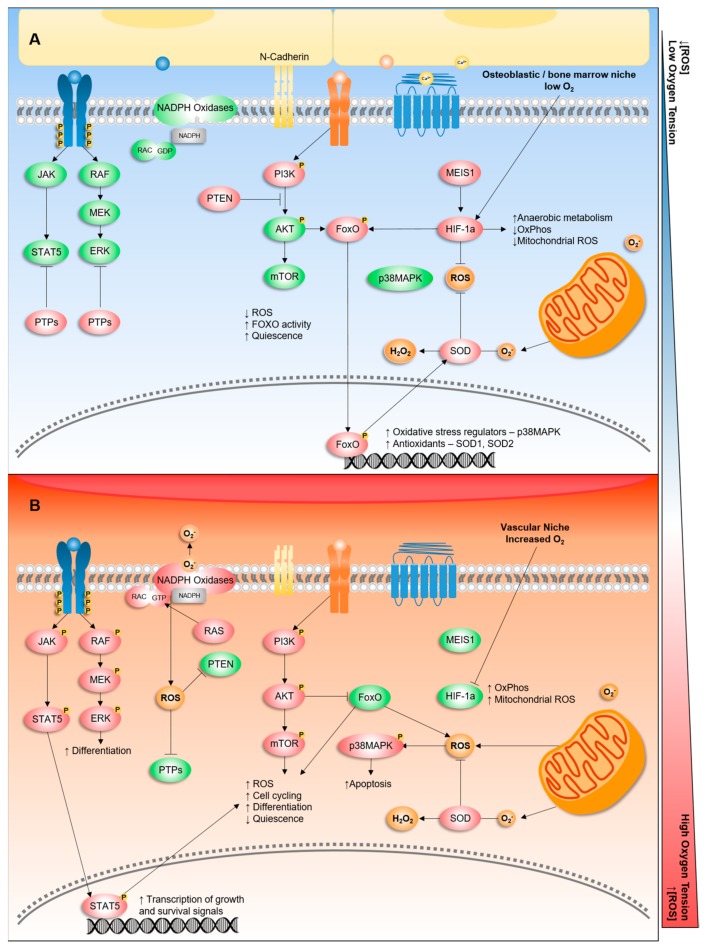
ROS-regulated haematopoietic stem cell (HSC) self-renewal and differentiation. (**A**) Within the low oxygen osteoblastic or bone marrow niche, anaerobic metabolism drives HIF1 and FOXO transcription to maintain quiescence and HSC self-renewal. (**B**) Following HSC release from the low oxygen osteoblastic or bone marrow niche to the oxygenated vascular niche, oxygen drives the activity of the NADPH oxidases, increasing ROS levels and promoting second messenger signalling, which in turn contributes to HSC growth, proliferation, and differentiation. Red = increased activity or expression. Green = decrease activity or expression. Blue = somatic mutation. Abbreviations Ox = cysteine oxidation, P = phosphorylation, Ca^2+^ = Calcium.

**Figure 3 ijms-20-06003-f003:**
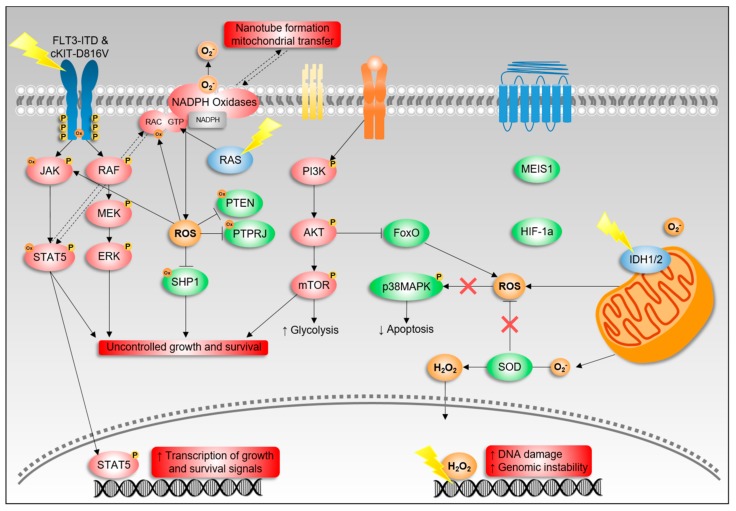
The role of ROS in driving oncogenic signalling in acute myeloid leukaemia (AML). Recurring somatic mutations to *FLT3*, *cKIT*, *RAS*, and *IDH1/2* drive intracellular ROS production in AML. High-level ROS production from NADPH oxidases drives second messenger signalling, through activation of kinases and the inactivation of PTPS, increased FLT3 signalling, and increased lipid peroxidation and genomic instability leading to chemotherapy treatment resistance. Red = increased activity or expression. Green = decrease activity or expression. Blue = somatic mutation. Abbreviations: PTP = protein tyrosine phosphatases, Ox = cysteine oxidation, P = phosphorylation.

**Table 1 ijms-20-06003-t001:** Proteins regulated by ROS-induced cysteine oxidisation.

Protein	Function	Activation/Inhibition	Disease/Cell	Assay
DUSP1, DUSP6, DUSP10, DUSP16	Phosphatase	Inhibition	Fibroblasts and HeLa Cells [97]	Electrophoretic mobility shift and phosphorylation screen
EGFR	Kinase	Activation	Breast Cancer [98] Lung Cancer [99,100]	2-Thiodimedone-specific IgG; selective, cell-permeable probe for detecting sulfenic acid
FLT3, FLT3-ITD	Kinase	Activation	AML cell lines [101]	Site-directed mutagenesis coupled with immunoprecipitation
JAK2, JAK3	Kinase	Inhibition	Pro-B Cells [102] Pancreatic β-Islet Cell [103]	Autokinase, in situ autophosphorylation, and transphosphorylation assays
PP2A-C	Phosphatase	Inhibition	Epithelial colorectal adenocarcinoma cells [95]	Immunoprecipitation coupled to antibody-based detection methods
PP2A-B56δ	Phosphatase	Inhibition	Jurkat cells and clinical human lymphomas [96]	Coimmunoprecipitation coupled to site-directed mutagenesis
PTPRJ	Phosphatase	Inhibition	AML [79]	Antibody-based detection methods
SHP1	Phosphatase	Inhibition	Fibroblasts [104] AML [105]	Antibody-based detection methods Immunoprecipitation coupled to phosphotyrosine screening
SHP2	Phosphatase	Inhibition	AML [105]	Immunoprecipitation coupled to phosphotyrosine screening
SRC	Kinase	Activation	Platelets [106] Fibroblasts [107]	Phosphotyrosine assessment, immunoprecipitation, biotinylation, and antibody-based detection methods

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
