# Peer review of "The Role of Reactive Oxygen Species in Acute Myeloid Leukaemia"

_ijms, 2019, doi:10.3390/ijms20236003_

Round 1

Reviewer 1 Report

The article by Sillar et al, “The Role of ROS in Acute Myeloid Leukemia” is a timely, well written and comprehensive review exploring both the positive and negative influences of ROS in haematopoietic differentiation and AML. The article is well structured, initially giving an overview of ROS and the antioxidants systems within cells to a more focused review of the literature in the context of haematopoiesis and AML. I have only some minor comments and suggestions that the authors may wish to incorporate into the review.

Lines 137-140. The authors mention the 1-Cys peroxiredoxin Prdx6, it is not clear why they focus on this particular peroxiredoxin as it has a number of activities apart from its function as a peroxidase e.g. phospholipase activity. Why are the typical 2-Cys peroxiredoxins (Prdx1-4) or the atypical Prdx5 not mentioned? The line about “the pool of reduced GSH is maintained by the reduction of GSSG by NADPH as an electron donor”, should the authors also mention glutathione reductase or are they suggesting that the catalytic cycle of the 1-Cys peroxiredoxin requires GSH for reduction of its peroxidatic Cys residue which has been shown in yeast. To date it is thought GSH plays this role for 1-Cys Prdx6 but has not been demonstrated and the literature would suggest GST pi performs this resolving function.

Similarly for completeness it might be worthwhile including the nitric oxide synthases in the introduction regarding endogenous generation of NO, as reactive nitrogen species are mentioned (line 105). Is there any evidence for peroxynitrite in AML?

Section 3 is very informative about the low/high ROS environment during haematopoietic differentiation which is suggested to be regulated by NOX enzymes. Generally I think the text could reference figure 2 on more occasions to aid in their text description. As the figures are to a high standard, I believe they could help in our understanding of the text.

Section 4 and the role of ROS in AML. From a purely personal perspective I would like to see if there is any evidence for ROS induced redox modifications on any of the regulatory proteins the authors describe. For instance formation of dimers/tetramers via disulphide bridges as a result of the changing redox environment during differentiation. The authors touch on this in sections of the review in relation to FLT3 and IDH2 but concentrate on mutated versions of the proteins as opposed to redox modifications that could affect the protein function. Do the mutated versions have altered redox sensitivity? This could also be significant in the differentiation pathway from low to high ROS as a result of increased NOX activation, what effects would the altered redox environment have on the activity of specific redox sensitive proteins? I appreciate that the authors make reference to JAK-STAT activation but is there any alterations in activity of Nrf2, NfkB and Hif1alpha.  In a similar vein and considering the highly susceptible redox sensitivity of metabolic enzymes such as IDH2, would it be worthwhile to consider briefly including some discussion in relation to metabolic flux that is highly dependent on the redox environment and whether this is altered in AML? Are there changes in glycolysis, oxidative phosphorylation or even pentose phosphate pathway activation to supply NADPH?

For the above reason I am a little confused by lines 464-468, “Drugs inhibiting oxidative phosphorylation, such as metformin [125, 126] and tigecycline [127, 128], have been explored in preclinical models and human trials of AML without any great success to date; however, realistically it would be hard to argue that any benefit seen from drugs targeting oxidative phosphorylation were due to decreased mitochondrial ROS synthesis.” Could this be clarified?

Would consider rewording line 481-482, “Firstly by causing DNA damage, which leads to genomic instability…..” Is there evidence for ROS causing specific mutations in DNA? The concentrations to reach nucleus and react with DNA would need to be extremely high.

Generally these are all minor comments and overall this is an excellent review.

Author Response

The article by Sillar et al, “The Role of ROS in Acute Myeloid Leukemia” is a timely, well written and comprehensive review exploring both the positive and negative influences of ROS in haematopoietic differentiation and AML. The article is well structured, initially giving an overview of ROS and the antioxidants systems within cells to a more focused review of the literature in the context of haematopoiesis and AML. I have only some minor comments and suggestions that the authors may wish to incorporate into the review.
a. Thanks to Reviewer 1 for these encouraging comments.

2. Lines 137-140. The authors mention the 1-Cys peroxiredoxin Prdx6, it is not clear why they focus on this particular peroxiredoxin as it has a number of activities apart from its function as a peroxidase e.g. phospholipase activity. Why are the typical 2-Cys peroxiredoxins (Prdx1-4) or the atypical Prdx5 not mentioned?
a. We have now simplified the text to reflect the peroxiredoxin family rather than just PRDX6 as details on all peroxiredoxin subtypes would be cumbersome reading.

3. The line about “the pool of reduced GSH is maintained by the reduction of GSSG by NADPH as an electron donor”, should the authors also mention glutathione reductase or are they suggesting that the catalytic cycle of the 1-Cys peroxiredoxin requires GSH for reduction of its peroxidatic Cys residue which has been shown in yeast. To date it is thought GSH plays this role for 1-Cys Prdx6 but has not been demonstrated and the literature would suggest GST pi performs this resolving function.
a. We have simplify the complexity of our statement which simply now states that “the pool of GSH is maintained by GSH peroxidase (GPx)-catalysed reactions which reduce ROS to oxidised GSSG, which in turn are reduced back to GSH by GSSG reductase at the expense of NADPH”.

4. Similarly for completeness it might be worthwhile including the nitric oxide synthases in the introduction regarding endogenous generation of NO, as reactive nitrogen species are mentioned (line 105). Is there any evidence for peroxynitrite in AML?
a. We thank the reviewer for this excellent suggestion and have added these data to lines 73-78

5. Section 3 is very informative about the low/high ROS environment during haematopoietic differentiation which is suggested to be regulated by NOX enzymes. Generally I think the text could reference figure 2 on more occasions to aid in their text description. As the figures are to a high standard, I believe they could help in our understanding of the text.a. We thank the reviewer for their compliment and wholeheartedly agree; we have added more references to Figure 2, where appropriate

6. For this comment, I have separated each subpoint into parts and addressed individually to be comprehensive.
Section 4 and the role of ROS in AML. From a purely personal perspective I would like to see if there is any evidence for ROS induced redox modifications on any of the regulatory proteins the authors describe.
6.1 For instance formation of dimers/tetramers via disulphide bridges as a result of the changing redox environment during differentiation. The authors touch on this in sections of the review in relation to FLT3 and IDH2 but concentrate on mutated versions of the proteins as opposed to redox modifications that could affect the protein function. Do the mutated versions have altered redox sensitivity? This could also be significant in the differentiation pathway from low to high ROS as a result of increased NOX activation, what effects would the altered redox environment have on the activity of specific redox sensitive proteins?
a. We thank the reviewer for their interesting perspective. ROS induced redox modifications leading to formation of dimers/tetramers via disulphide bridges, as a result of the changing redox environment, leads to the inter-protein disulphide bond formation for Peroxiredoxins (PMID: 15031298). However, oxidation of PTPs for the most part, leads initially to the formation of sulfenic acid, followed by the formation of reversible oxidation products, which differ among different PTPs. Disulfide formation occurs with a nearby resolving cysteine particularly for SHP1/2, PTEN, and Cdc25 (PMID: 23861395).
b. In addition, a paper accepted for publication from the Böhmer lab on the 8th of September 2019 showed that under situations of oxidative stress, FLT3-ITD is oxidised at cytoplasmic cysteine residues 695 and 790 and go on to elegantly show that cysteine 790 is responsible for FLT3 activation. We have added these additional data to the manuscript text, table 1 and figure 3.
6.2 I appreciate that the authors make reference to JAK-STAT activation but is there any alterations in activity of Nrf2, NfkB and Hif1alpha
a. We have added an additional reference to line 401 which summarises these oxidation events.
6.3 In a similar vein and considering the highly susceptible redox sensitivity of metabolic enzymes such as IDH2, would it be worthwhile to consider briefly including some discussion in relation to metabolic flux that is highly dependent on the redox environment and whether this is altered in AML? Are there changes in glycolysis, oxidative phosphorylation or even pentose phosphate pathway activation to supply NADPH
a. Metabolic abnormalities are multiple and heterogeneous among AML patients. Reports frequently suggest that glycolysis is upregulated to which AML cells become dependent implicating the activity of the PPP. Commensurate are reports of deregulation of oxidative phosphorylation in AML cells, shown to play critical roles in leukaemogenesis, both of patients at diagnosis and following chemoresistance. Furthermore, the high plasticity of AML cells enables them to adapt their metabolism and ensure their survival even under selective inhibition of some oncogenic signalling pathways. Indeed NADPH plays an important role in homeostasis and the survival of cancer cells under metabolic stress limiting therapeutic intervention through their modulation of ROS and metabolic pathways. We thank the reviewer for this interesting perspective and agree that a full description of the links between oxidative stress and metabolism is a very worthy pursuit. However, at the current time we feel this is beyond the scope of this review.

7. For the above reason I am a little confused by lines 464-468, “Drugs inhibiting oxidative phosphorylation, such as metformin [125, 126] and tigecycline [127, 128], have been explored in preclinical models and human trials of AML without any great success to date; however, realistically it would be hard to argue that any benefit seen from drugs targeting oxidative phosphorylation were due to decreased mitochondrial ROS synthesis.” Could this be clarified?a. We have modified the text to suggest the failure of drugs targeting oxidative phosphorylation in human trials of AML, as the data would suggest that the NOX family are the primary drivers of ROS production in AML, which may partly explain this.

8. Would consider rewording line 481-482, “Firstly by causing DNA damage, which leads to genomic instability…..” Is there evidence for ROS causing specific mutations in DNA? The concentrations to reach nucleus and react with DNA would need to be extremely high.
a. We have modified the text to remove confusion.

9. Generally these are all minor comments and overall this is an excellent review.
a. Thank you again to Reviewer 1 for their expert comments.

Reviewer 2 Report

In this review article, Sillar et al summarise the role of reactive oxygen species (ROS) in normal haematopoiesis and also in acute myeloid leukaemia (AML). They discuss the outcomes of AML and the fact that many patients die due to a lack of successful treatments leading to a very poor overall survival rate. The authors state that ROS have been shown to promote leukemogenesis in AML along with the regulation of normal haematopoiesis. At the same time, ROS can stimulate genomic instability caused by damaged DNA leading to chemotherapy resistance. The authors evaluate the role of ROS in both normal haematopoiesis and also in AML.

This review assesses the roles that ROS play in the pathogenesis of AML with a main focus on current and future therapeutic interventions based on the fact that modulating ROS levels may play a key role.

Comments: The authors highlight the issues relating to AML as an aggressive cancer leading to bone marrow failure and death of an individual. They state that there has been a very small increase in survival over the last 40 years, mainly due to supportive/palliative care and they highlight the fact that most progress has been made in the under 40 age group where survival times can extend to 5 years in 50% of patients. Sadly, patients over the age of 70 have a less than 5% chance of long-term survival. Treatment for AML includes small molecule inhibition and agonist therapies based on the 1970’s treatment plans. The authors stress the fact that relapse rates are high and follow up appointments are frequent. It is now well recognised that ROS play important roles in both physiological and pathological cell signalling. A low ROS environment helps to maintain haemopoietic stem cell (HSC) quiescence, self-renewal and long-term survival. The authors cover the main areas of interest in this review and assess topics in a reasonably robust manner. They highlight the importance of ROS in maintaining HSC quiescence and discuss the evidence of expression of ‘NOX family’ members in myeloid cells including the functional consequences in knockout models and patients.

The authors review the state of the literature relating to ROS in AML and discuss papers that demonstrate that ROS are elevated in cancer cells.

The supporting diagrams in this review are helpful and informative and the text is easy to read.

Can the authors confirm whether the figure they refer to on line 432 is actually correct? The same applies to the figure on line 436. Is this correct?

There are a few basic typing errors here and there (such as lines 38, 78, 82, 284, 479) with words in the wrong places. These just need a bit of tidying up.

The authors state there is conclusive evidence now showing that ROS are elevated in AML and that there are most likely numerous sources of ROS. These include the mitochondria, NOX complex and other metabolic sources. However, the authors conclude that most of the evidence to date supports the NOX complex (in particular NOX2) as the primary driver of ROS production. It is important to note that ROS scavengers and antioxidants have not been found to prevent or treat cancer and so there is no evidence to support this type of approach in AML.

The review itself covers many relevant previous publications and discusses the limitations and contradictions within those papers. It is well written and it does what it says in the title.

Author Response

In this review article, Sillar et al summarise the role of reactive oxygen species (ROS) in normal haematopoiesis and also in acute myeloid leukaemia (AML). They discuss the outcomes of AML and the fact that many patients die due to a lack of successful treatments leading to a very poor overall survival rate. The authors state that ROS have been shown to promote leukemogenesis in AML along with the regulation of normal haematopoiesis. At the same time, ROS can stimulate genomic instability caused by damaged DNA leading to chemotherapy resistance. The authors evaluate the role of ROS in both normal haematopoiesis and also in AML. This review assesses the roles that ROS play in the pathogenesis of AML with a main focus on current and future therapeutic interventions based on the fact that modulating ROS levels may play a key role.
The authors highlight the issues relating to AML as an aggressive cancer leading to bone marrow failure and death of an individual. They state that there has been a very small increase in survival over the last 40 years, mainly due to supportive/palliative care and they highlight the fact that most progress has been made in the under 40 age group where survival times can extend to 5 years in 50% of patients. Sadly, patients over the age of 70 have a less than 5% chance of long-term survival. Treatment for AML includes small molecule inhibition and agonist therapies based on the 1970’s treatment plans. The authors stress the fact that relapse rates are high and follow up appointments are frequent. It is now well recognised that ROS play important roles in both physiological and pathological cell signalling. A low ROS environment helps to maintain haemopoietic stem cell (HSC) quiescence, self-renewal and long-term survival. The authors cover the main areas of interest in this review and assess topics in a reasonably robust manner. They highlight the importance of ROS in maintaining HSC quiescence and discuss the evidence of expression of ‘NOX family’ members in myeloid cells including the functional consequences in knockout models and patients.The authors review the state of the literature relating to ROS in AML and discuss papers that demonstrate that ROS are elevated in cancer cells.

1. The supporting diagrams in this review are helpful and informative and the text is easy to read.
a. We thank Reviewer 2 for their comments

2. Can the authors confirm whether the figure they refer to on line 432 is actually correct?
a. This this correct. Nanotubule formation is included on Figure 3 and referred to in text.

3. The same applies to the figure on line 436. Is this correct?
a. We thank Reviewer 2 for picking this up. The reference to figure 2 in this instance is not relevant

4. There are a few basic typing errors here and there (such as lines 38, 78, 82, 284, 479) with words in the wrong places. These just need a bit of tidying up.
a. We thank Reviewer 2 for picking these up and have corrected as appropriate

5. The authors state there is conclusive evidence now showing that ROS are elevated in AML and that there are most likely numerous sources of ROS. These include the mitochondria, NOX complex and other metabolic sources. However, the authors conclude that most of the evidence to date supports the NOX complex (in particular NOX2) as the primary driver of ROS production. It is important to note that ROS scavengers and antioxidants have not been found to prevent or treat cancer and so there is no evidence to support this type of approach in AML.
a. We agree and thank Reviewer 2 for highlighting this important point. As the reviewer and we state, unfortunately approaches to reduce ROS have not improved outcomes for AML patients. However, monotherapeutic strategies to treat aggressive cancers rarely work. Therefore we believe, that future studies and clinical trials will incorporate drugs targeting the machinery responsible for ROS production (which in no doubt modulates intracellular oxidative stress, driving second messenger signalling - as we have described) in combination with drugs targeting oncogenic signalling, which may enhance their therapeutic efficacy and durability. Then we may see increased efficacy of precision therapies to treat highly aggressive cancers like AML.

6. The review itself covers many relevant previous publications and discusses the limitations and contradictions within those papers. It is well written and it does what it says in the title.
a. We thanks Reviewer 2 for their expert comments.